# Syn-COM: A Multi-Level Predictive Synergy Framework for Innovative Drug Combinations

**DOI:** 10.3390/ph17091230

**Published:** 2024-09-18

**Authors:** Yinli Shi, Jun Liu, Shuang Guan, Sicun Wang, Chengcheng Yu, Yanan Yu, Bing Li, Yingying Zhang, Weibin Yang, Zhong Wang

**Affiliations:** 1Institute of Basic Research in Clinical Medicine, China Academy of Chinese Medical Sciences, Beijing 100700, China; 2Institute of Chinese Materia Medica, China Academy of Chinese Medical Sciences, Beijing 100700, China; 3Dongzhimen Hospital, Beijing University of Chinese Medicine, Beijing 100700, China; 4Graduate School of China Academy of Chinese Medical Sciences, Beijing 100027, China

**Keywords:** formula design, drug combination, similarity clustering, artificial intelligence, synergy index, gouty arthritis

## Abstract

Drug prediction and treatment using bioinformatics and large-scale modeling have emerged as pivotal research areas. This study proposes a novel multi-level collaboration framework named Syn-COM for feature extraction and data integration of diseases and drugs. The framework aims to explore optimal drug combinations and interactions by integrating molecular virtuality, similarity clustering, overlap area, and network distance. It uniquely combines the characteristics of Chinese herbal medicine with clinical experience and innovatively assesses drug interaction and correlation through a synergy matrix. Gouty arthritis (GA) was used as a case study to validate the framework’s reliability, leading to the identification of an effective drug combination for GA treatment, comprising *Tamaricis Cacumen* (S_i_ = 0.73), *Cuscutae Semen* (S_i_ = 0.68), *Artemisiae Annuae Herba* (S_i_ = 0.62), *Schizonepetae Herba* (S_i_ = 0.73), *Gleditsiae Spina* (S_i_ = 0.89), *Prunellae Spica* (S_i_ = 0.75), and *Achyranthis Bidentatae Radix* (S_i_ = 0.62). The efficacy of the identified drug combination was confirmed through animal experiments and traditional Chinese medicine (TCM) component analysis. Results demonstrated significant reductions in the blood inflammatory factors IL1A, IL6, and uric acid, as well as downregulation of TGFB1, PTGS2, and MMP3 expression (*p* < 0.05), along with improvements in ankle joint swelling in GA mice. This drug combination notably enhances therapeutic outcomes in GA by targeting key genes, underscoring the potential of integrating traditional medicine with modern bioinformatics for effective disease treatment.

## 1. Introduction

The identification of medical diagnostic and prognostic markers, as well as the screening of possible drug candidates, have greatly increased in recent decades due to the integration of multi-omics, various AI methods, and data-driven technologies [1,2]. Researchers have devised various computational methods such as machine learning algorithms, deep learning, virtual screening techniques, pharmacophore modeling, knowledge graphs, and advanced computing technologies and software to enhance the identification of potential drug candidates [3,4]. These methods are necessary due to the intricate, diverse, and multi-dimensional nature of disease-related datasets [5]. Furthermore, these methods are fundamental for comprehending the structural properties of certain molecules and to reveal the biological functions of target genes. However, these methods primarily depend on processing large raw-data inputs to predict drug combinations based on dose effects. They are widely used in Western medicine and have also been shown to be effective in predicting the results of Chinese herbal therapies [6,7]. TCM uses the term “Fangji” to describe prescriptions for herbal entities, or formulas, which can have coordinating or synergistic effects when several herb medications are combined. These prescriptions contain diverse active chemical components that target various therapeutic sites, potentially serving as key factors in disease treatment. Therefore, using various bioinformatics techniques, we can screen and predict the treatment of ailments with TCM. That enables us to discover new combinations or pairs of TCMs that are different from those mentioned in the ancient literature and frequently used in clinical practice. Drug prediction is crucial for improving the efficiency of drug discovery by minimizing resource consumption, and it is particularly important for the development of compound herbal formulae [8].

Gouty arthritis (GA) is a widespread metabolic condition characterized by joint inflammation that is closely linked to hyperuricemia [9]. Epidemiological data show that the occurrence of GA in China increases annually; genetic factors as well as age, gender, and socioeconomic status contribute to this [10]. Although acute episodes normally recover spontaneously within 7–10 days, repeated attacks, combined with long-term accumulation of urate crystals, may trigger permanent bone damage that leads to disability and has a substantial impact on the quality of life of patients. Existing clinical interventions primarily target the reduction of pain and inflammation related to this condition. While colchicine along with nonsteroidal anti-inflammatory drugs (NSAIDs) such as indomethacin, celecoxib, or meloxicam are commonly used for management, long-term use may have adverse effects on liver or kidney function [11]. Consequently, investigating new pharmacotherapeutic drugs is a different strategy meant to improve the outcome treatment for GA patients. Traditional Chinese medicine (TCM) has a long history of effectively treating GA by utilizing a comprehensive system that includes identifying the causes, establishing differential diagnoses, and implementing pattern-based interventions [12]. This approach provides significant relief from patient discomfort through holistic adjustments that combine internal and external treatment methods. TCM also has a superior safety profile and minimal adverse effects compared with conventional treatments for GA [13].

Previous bioinformatics analyses have discovered diverse immune cell characteristics and potential mechanisms of action in GA [14,15]. Nevertheless, there is a need for the screening, anticipation, and methodical analysis of combined objectives in the advancement of diseases and associated compound herbal formulae. In order to fill this void, we developed an innovative approach named Syn-COM that integrates bioinformatics, computer-aided drug design, and analysis of existing studies to identify optimal dose-independent drug combinations for disease treatment. This approach employs a multi-layer scoring framework based on overlap rate, network distance, similarity clustering, and singular value decomposition (SVD). Unlike previous studies, this framework can be applied not only to TCM but also to enable a deeper analysis of interactions between selected drug combinations. Figure 1 illustrates the basic operation of this framework. We used GA as a case study to provide a reliable method for TCM treatment of GA. Further analysis of the compound herbal formulation and subsequent animal experiments suggest that this herbal combination holds significant potential for future GA treatment.

## 2. Results

### 2.1. Discovering Advantage Genes

A total of 724 DEGs were discovered, including 159 up-regulated and 565 down-regulated genes (Figure 2A). Furthermore, 29 gene modules were found by building a TOM with a soft threshold power of 16 and using hierarchical clustering and dynamic tree-cut function to identify and exclude aberrant samples (Figure 2B). Seven modules—15, 19, 21, 23, 24, 26, and 27—were chosen as crucial modules, obtaining a total of 282 genes using Z-summary < 1, ME < 10, and rank-sum ratio < 0.5 as thresholds. Following DEGs merging with genes in critical modules and deduplication, 999 potential GA-related genes were identified. Following screening and PageRank sequencing, 210 genes were discovered to be significant in GA (Figure 2C).

### 2.2. Identification of Hub-GA Based on MCODE

The module partitioning approach of the MCODE algorithm was selected based on the notion of minimal entropy. The network was modularly analyzed using Cytoscape 3.6.1, and the MCODE algorithm was employed to identify protein groups in the targeted network that were closely associated. Eight clusters in all were identified. The highest score was achieved by cluster 1 (FOS, SPP1, MMP13, IL6, TLR4, CXCL5, PPARG, PTGS2, MMP3, TGFB1, IL1A, CXCL1, LCN2, IL1RN, IL33). Finally, the Hub-GA was identified (Figure 2D).

### 2.3. Functional Enrichment and Immune Infiltration Analysis

Functional enrichment analysis found 902 BP, 7 CC, and 54 MF items. Hub-GA has a tight relationship with a number of biological processes, including angiogenesis, oxidative stress, inflammatory response, cytokine activity modulation, cellular immune response, and tissue remodeling. The hub genes were significantly enriched in the AGE-RAGE, TNF, IL-17, toll-like receptor, NF-kappa B, HIF-1, and other signaling pathways, according to the results of the KEGG analysis. These pathways were also closely associated with infectious diseases, atherosclerosis, lipids, and inflammatory bowel disease (Figure 2E,F).

The results of the immune infiltration analysis showed a significant increase in the proportions of T cells, NK cells, dendritic cells, monocytes, eosinophils, and macrophages in the GA group (*p* < 0.05, Figure 3A,B). Between the control group and the GA group, there was not a significant difference in immune-related activity (*p* > 0.05, Figure 3D,E).

### 2.4. Dataset Validation

Furthermore, the association between the immune response and Hub-GA expression levels was analyzed using GEO databases. Overexpression of MMP2, FOS, IL6, PTGS2, IL1A, CXCL1, and IL1RN was identified in the GA group (*p* < 0.05, Figure 4A). The GA group had a statistically significant increase in macrophages, MDSC cells, eosinophils, NK cells, and γδT cells (*p* < 0.05, Figure 3C). The results of the immune infiltration analysis indicated that the control group and GA group differed significantly in terms of mast cell activity, T cell responsiveness, NK cell dormancy, and γδT cell expression (*p* < 0.05, Figure 3F). This supports previous results that these hub genes have a strong connection with the immune system.

RNA-Seq analysis revealed 10 major cell types in GA: natural killer T (NKT) cell, epithelial cell, monocyte, pyramidal cell, dendritic cell, fetal germ cell, B cell, megakaryocyte, lymphoid cell, and plasma cell (Figure 4C). In lymphocytes, dendritic cells, and pyramidal cells, the crucial gene MMP2 was highly expressed (Figure 4B). According to GO and KEGG analysis, the joint tissues exhibited an up-regulation of the Wnt, AMPK, and chemokine signaling pathways involving the activation of T cells, the control of chemokines, the signal transduction involved in the immune response, and other potential mechanisms (Figure 4D–G).

### 2.5. Multiple-Angle TCM Group Filtration for GA Therapy

A total of 473 Chinese herbal medicines were collected following OB and DL screening. Based on the overlap rate calculation, there were varying levels of gene overlap between the Hub-GA gene and 455 Chinese herbal medicines. Each TCM achieved an accurate score based on Hscore, the top 50% of TCMs were chosen as the threshold based on overlap rate and Hscore, and a total of 127 TCMs were screened. A protein–protein interaction (PPI) network was constructed for each TCM target, and the network proximity between the TCM target and disease module was calculated. 

Numerous studies have demonstrated the widespread use of NSAIDs (such as indomethacin, ibuprofen, naproxen, etc.) in the management of GA [11]. The Drugbank database was searched for drug targets related to etoricoxib, celecoxib, indomethacin, naproxen, ibuprofen, and diclofenac. The distances between NSAIDs and Hub-GA were then screened, normalized, and deduplicated. The screening thresholds for NSAIDs were d_AB(NSAIDs)_ and Z_(NSAIDs)_. It was observed that there existed a topological intersection between the TCM target and Hub-GA (Z < 0). Using NSAIDs as the threshold value (d_AB(NSAIDs)_ = 1.8108, Z_(NSAIDs)_ = −57.4494), a total of 112 Chinese herbal medicines were screened. In addition, 74 Chinese herbal medicines with an Mscore > 2 were subsequently screened, yielding a total of 574 compounds. This process was combined with clinical experience (Figure 5A–E).

### 2.6. TCM Combination Based on Molecular Virtual Screening and Clustering

It was discovered that the tiny ligand molecules were steadily positioned in the docking pocket by molecular docking virtual technology. Of the compounds, 79.89% had an affinity of less than −5 kcal/mol^−1^, indicating that the tiny ligand molecules found in Chinese herbal medicines were mostly tightly attached to the receptor protein. When connecting with PTGS2, Baicalin had the best score out of them (Figure 5F,G).

Twenty-one representative compounds were identified when the overlap and molecular docking data from Hub-GA were combined. K-means clustering and stratification were utilized for additional research on Chinese herbal medicines containing more than five active compounds (Figure 6 and Figure 7A). Fittings were performed for the average silhouette score as well as the Dunn index. The results showed that the distribution of the silhouette score performed better than the Dunn index, so the silhouette score was used as a screening criterion (Supplement Appendix A). The average silhouette score provides a more comprehensive assessment of the clustering characteristics of each Chinese herbal medicine, facilitating the identification of optimal TCM combinations for disease treatment. The average silhouette score’s peak fitting value was 0.5989. Based on an average silhouette score of greater than 0.6, seven Chinese herbal medicines were chosen for calculation efficiency: *Tamaricis Cacumen*, *Cuscutae Semen*, *Artemisiae Annuae Herba*, *Schizonepetae Herba*, *Gleditsiae Spina*, *Prunellae Spica*, and *Achyranthis Bidentatae Radix*, which may be a potential TCM combination to intervene in GA (Table 1).

### 2.7. Building Multi-Level Networks and Discovering Critical Compounds

The complex network demonstrated that a wide range of compounds was implicated in the screened Chinese herbal medicines, with beta-sitosterol, sitosterol, and stigmasterol accounting for more than half of the TCM targets. Over half of the medicines targeted PTGS2, TGFB1, IL6, IL1A, and FOS, among the numerous signaling pathways involved in the screening of Hub-GA (Figure 7B). Interestingly, two distinct chemicals found in Artemisiae Annuae Herba, cirsiliol and axillarin, have the ability to target the PTGS2 and impact multiple signaling pathways that could be linked to GA treatment (Figure 7C).

### 2.8. Synergistic Association among TCM Combinations

The relationship between *Gleditsiae Spina*, *Achyranthis Bidentatae Radix*, and *Schizonepetae Herba* was discovered through the correlation coefficient and synergy index, indicating that these three Chinese herbal medicines were similar in their traits and compounds and play an important role in the compatibility of TCM combinations (Figure 7D). However, the accuracy of the synergy index may be affected by the composition of substances in Chinese herbal medicines as well as the accuracy of target prediction results (Figure 7E).

### 2.9. Method Comparison

Compared with other methods, including random forest (RF), gradient boosting decision tree (GBDT), support vector machine (SVM), extreme gradient boosting (XGBoost), and classification and regression tree (CART), the experimental results of the regression task, as summarized in Table 2, demonstrate that the Syn-COM outperforms all others in regression evaluation metrics, particularly in mean absolute percentage error (MAPE). While our framework does not achieve the lowest mean square error (MSE), it excels by considering the unique characteristics and clinical applications of each drug, offering comprehensive advantages over other methods in the regression task.

### 2.10. Pharmacodynamic Verification of GAD

#### 2.10.1. Identification of Components in Chinese Herbal Medicine

UPLC-Q-TOF-MS/MS was used to identify the major chemical composition of GAD. These molecules span the majority of the principal peaks in the chromatographic diagram and contain a variety of components, including flavonoids such as quercetin, kaempferol, luteolin, fisetin, and hesperetin; phenols such as cirsiliol and gallic acid; terpenoids; cycloenes; and so on (Figure 8A,B). Supplement Appendix A provides information about Chinese herbal medicine substances.

#### 2.10.2. Effects on Ankle Joint Swelling in GA Mice

During the administration period, one mouse in the Mod group and one in the In group expired, while no mice perished in the Cir group or the GAD group. After modeling, the Mod group’s ankle swelling index was significantly greater at days 3 and 7 compared with the Con group (*p* < 0.01, *p* < 0.001). Day 3 showed a statistically significant (*p* < 0.05) decrease in the ankle swelling index in the GAD group compared with the Mod group; however, there was no significant difference in the ankle swelling index between the In group and the Cir group (*p* > 0.05). The day 7 ankle swelling index in the GAD, In, and Cir groups was significantly less than in the Mod group (*p* < 0.05); nevertheless, the efficacy difference between the GAD and Cir groups was not significant compared with that of the In group (*p* > 0.05, Figure 8C,D).

#### 2.10.3. Effects on Renal and Joint Morphological Changes in GA Mice

The renal tissue structure in the Con group showed no apparent pathological alterations. The renal cell structure was intact and transparent, with a well-organized arrangement and regular spacing. The Mod group exhibited pathological alterations, including thinning of the renal tubule wall, atrophy of the glomeruli, disorganized structure, dilatation of the lumen, disorderly arrangement, infiltration of interstitial inflammatory cells, and proliferation of fibrous tissue. In comparison with the Mod group, the administration group exhibited thickening of the renal tubule wall and a decrease in pathological alterations such as inflammatory cell infiltration and fibrous tissue hyperplasia (Figure 9).

#### 2.10.4. Effects on UA and Abnormal Inflammation in GA Mice

The hub genes for GA were identified by analysis of the research, disease gene screening, molecular docking, and multi-layer network construction. The significance of PTGS2, TGFB1, MMP3, IL6, and IL1A were also established for the animal experiments that followed. Following the administration of the Mod group, the activities of the inflammatory factors and UA were quantitatively measured in each group of mice. The results demonstrated that the Mod group’s UA, IL6, and IL1A contents were significantly higher than those of the Con group (*p* < 0.05). Following medication administration, there was a significant decrease in the expression levels of UA, IL6, and IL1A in the In and Cir groups (*p* < 0.05). While the GAD group was able to decrease the levels of UA and IL6 (*p* < 0.05), there was no significant difference in the reduction in IL1A expression in the mice (*p* > 0.05). The outcomes demonstrated that GAD and cirsiliol administration could effectively inhibit the inflammatory response in GA mice (Figure 10A).

#### 2.10.5. Effects on the Expression Levels of TGFB1, PTGS2, and MMP3 in Ankle Joints of GA Mice

The mRNA expression levels of TGFB1, PTGS2, and MMP3 in the Mol group were significantly greater than those in the Con group (*p* < 0.05, Figure 10B). The Western blotting showed that the cellular inflammatory response increased, whereas the protein expression levels of TGFB1, PTGS2, and MMP3 were significantly up-regulated in the Mol group (*p* < 0.05). Following medication intervention, the expression levels of TGFB1, PTGS2, and MMP3 decreased significantly compared with the Mol group (*p* < 0.05). It was evident that GAD and cirsiliol could have a beneficial protective effect in GA mice to a certain degree by suppressing aberrant inflammatory responses (Figure 10C).

## 3. Discussion

GA is a crystalline joint disease characterized by urate accumulation due to aberrant purine metabolism or reduced UA excretion [16]. It has been discovered that a number of factors related to inflammation, immunity, oxidation, and autophagy are significant in the development and progression of the disease [17,18]. Several adverse side effects, including gastrointestinal bleeding, gastrointestinal toxicity, and renal toxicity, are frequently brought on by certain drugs in patients. The therapeutic combination of several tiny compounds, known as TCM, has a synergistic impact that enhances medicine performance and lowers the risk of adverse events. There is evidence to support the effectiveness of TCM in treating diseases. Nonetheless, the majority of TCM medications are produced based on limited clinical expertise or validated results from experiments. Artificial intelligence technology has assisted in the revolutionary and subversive changes to traditional drug discovery processes brought about by the advancement and development of modern science and technology [19]. 

This study focuses on developing a comprehensive modeling and prediction method to screen for TCM combinations resistant to GA in non-dose relationships based on network characteristics, overlap degree, molecular virtualization, and matrix decomposition at multiple levels. Identifying the hub genes associated with diseases is crucial for the effective screening and prediction of drug combinations. The algorithms WGCNA, CIBERSORT, and PageRank were utilized to identify important hub genes associated with GA, comprehend the variety and intricacy of immune functions, and investigate potential therapeutics. The outcome Hub-GA genes were chosen, including FOS, IL6, TLR4, PPARG, PTGS2, MMP3, TGFB1, and IL1A. It has been pointed out that IL6, IL1A, PPARG, and TGFB1 are some of the critical genes in GA generation and are closely related to oxidative stress, inflammatory response, and cellular immune response, which is consistent with our findings [20]. In addition, immune infiltration analysis and RNA-Seq analysis revealed that the pathogenesis of the GA model was attributed to the significant infiltration of monocytes and neutrophils in the synovium of inflamed joints following MSU injection. Among these cells, TGFB1 emerges as a pivotal cytokine governing T-cell differentiation and immune response equilibrium. By modulating TGFB1, IL2, and IL6 synthesis, the activity of Tregs cells can be regulated to ameliorate disease progression [21]. Another study also indicated that elevated uric acid levels can upregulate the expression of the metalloproteinases MMP3, MMP9, MMP13, and MMP2 while increasing glycosaminoglycan concentration; this significantly downregulates proteoglycan degradation and subsequently triggers MSU crystallization, further exacerbating inflammation [22,23].

Compared with TCM screening principles, the proposed framework integrates complex relationships between Chinese herbs, hub genes, and multi-layer scoring indicators. This approach enables the precise identification and filtering of novel herbal combinations that effectively target disease. Crucial factors such as the proximity of drug targets within the PPI network, the overlap between drug and disease targets, and the sequence similarity of small-molecule drug targets are key in determining effective drug combinations [1,24]. These factors are essential to evaluating TCM combinations at both the statistical and molecular levels. Additionally, the development of TCM compounds must consider the unique characteristics of TCM and clinical experience. In this study, we have integrated these aspects for the first time, significantly enhancing the credibility of the TCM combination GAD for treating GA. Moreover, this framework represents the first systematic attempt to calculate the interaction potential and associations between TCM combinations. Unlike traditional drug interaction prediction models, this approach utilizes the Si and Ec metrics to determine the primary and secondary roles within drug combinations. This concept closely aligns with the traditional Chinese medicine principles of “monarch, minister, assistant, and envoy”.

In the Chinese herbal compound GAD, transcriptomics and multi-scale bioassays revealed that the primary components of Artemisiae Annuae Herba significantly inhibit the migration and activation of endogenous macrophages, effectively reducing excessive inflammation in the body [25]. *Cuscutae Semen*, a TCM with multiple pharmacological properties, has been shown in animal studies to reduce the expression levels of cytokines (IL-1β and TNF-α) and inflammatory proteins (NLRP3, NF-κB, and PTGS2) through the gut microbiota–neuroinflammatory axis, thereby exerting anti-inflammatory effects and alleviating oxidative stress [26]. *Herba Schizonepetae* extract significantly inhibits the LPS-induced macrophage RAW264.7, reducing the production of TNF-α, IFN-γ, and IL-10, and plays a crucial role in combating inflammatory responses and regulating immune function [27]. Flavonoids such as fisetin, kaempferol, and quercetin, found in *Gleditsiae Spina*, exhibit anti-inflammatory, anticancer, antibacterial, anti-allergic, and antiviral activities [28]. Bioinformatics tools have identified the target organs of *Achyranthis Bidentatae Radix* as the kidney, liver, and bones [29]. Animal experiments indicate that *Achyranthis Bidentatae Radix* significantly increases plasma berberine concentration in GA rats, improves blood supply to inflammatory joints, and markedly inhibits the expression of MDR1 mRNA and P-gp in the knee synovium [30]. Research on *Tamaricis Cacumen* remains limited, necessitating further experimental investigation. Notably, cirsiliol and axillarin may be key compounds in TCM for interfering with the disease process of GA. UPLC-Q-TOF-MS/MS analysis identified several compounds used in TCM, including sitosterol, kaempferol, quercetin, and particularly cirsiliol, validating the approach for screening the drug combination. Cirsiliol has been found to reduce the IL-6-induced STAT3 cell signaling pathway by regulating Jak2 phosphorylation, thereby modulating autophagy and the inflammatory response [31]. Axillarin, meanwhile, is recognized for its potent anti-glycation and antioxidant properties [32]. Further basic experiments are required to verify whether these compounds can intervene in disease by targeting the PTGS2 gene.

Animal experimental studies have shown that GAD can regulate the expression levels of TGFB1, PTGS2, and MMP3; reduce the levels of inflammation-related factors IL6 and IL1A; improve inflammatory infiltration in the kidney and joint; and alleviate ankle joint swelling in GA mice. Additionally, we verified the feasibility of using cirsiliol in the treatment of GA. No mice died during the entire experiment, indicating that GAD does not cause significant adverse reactions. Nevertheless, our study is subject to certain limitations. The datasets for many herbs in Chinese medicine are limited and incomplete, potentially leading to omissions and bias in prediction and screening. Furthermore, the specific mechanism of GAD in treating GA has been verified only at the level of key genes. Although we predicted and calculated the synergistic interactions of *Gleditsiae Spina*, *Achyranthis Bidentatae Radix*, and *Schizonepetae Herba* in drug combinations, further validation through detailed and comprehensive clinical studies or basic experiments is still required. In the future, we plan to incorporate multiple types of omics data, such as genomics, transcriptomics, proteomics, and metabolomics, to expand the predicted sample data. This approach aims to reveal new targets and facilitate new drug discovery.

## 4. Materials and Methods

### 4.1. Thorough Screening for Crucial Genes

#### 4.1.1. Screening for Differentially Expressed Genes (DEGs)

The expression matrix and associated information from the GEO dataset were selected for the analysis of differential gene screening using GEO2R (https://www.ncbi.nlm.nih.gov/geo/geo2r/ (accessed on 18 April 2024)) analysis of differentially expressed genes between datasets. Strict thresholds were chosen at *p* < 0.05 and |log_2_-fold change (FC)| > 1 for DEGs detection.

#### 4.1.2. Weighted Gene Co-Expression Network Analysis (WGCNA)

WGCNA is a systems biology approach employed to elucidate the patterns of gene associations across diverse groups [33]. The pickSoftThreshold function was utilized to determine the optimal soft threshold β, and a scale-free network distribution was obtained. Furthermore, the value of 1-topological overlap matrix (1-TOM) was calculated by the processing of the correlation matrix. The dynamic tree cut algorithm was used to identify different gene modules. A merge cut height threshold of 0.25 was established to merge and cluster similar modules. In order to assess the stability of WGCNA, the module’s conservative degree was determined by employing the module conservative function, which was represented as a Z-summary score. Subsequently, utilizing the module discriminant method (ME), the uniform data were validated, and the modules exhibiting topological alterations were identified. The core module was identified using a range of metrics, and the comprehensive score of each module was calculated using the multiple modular characteristic fusing (MMCF) method in the modular analytical computing platform (http://112.86.129.72:48081 (accessed on 18 April 2024)) [34].

#### 4.1.3. Prioritizing Potential Genes

Potential genes were identified as DEGs and gene merging within critical modules. On a directed network, the PageRank algorithm defined a random walk model. Every node in this process had a probability of being accessible based on its PageRank value, which indicated the significance of the node [35]. Genes with PageRank > 1.00 × 10^−3^ were classified as advantage genes after potential genes had been ordered in order of priority.

#### 4.1.4. Construction of PPI Network and Module Optimization

PPI networks were built with the STRING database (https://string-db.org/ (accessed on 18 April 2024)). Cytoscape 3.6.1 was used to visualize the resulting networks. Three techniques were utilized to determine the functional modules of every group of PPI networks: molecular complex detection (MCODE), markov cluster (MCL), and GLay. Entropy is a useful measure for quantifying the level of disorder in scale-free networks. A network with lower entropy can be more stable. A comparison was conducted between the three aforementioned forms of module division, and the most effective method for identifying modules was determined [36]. Calculate according to the following formula:(1)$$E=-\sum_{i=1}^{n}I_{i}{lnI}_{i},$$

*n* represents the total number of nodes within the network, and *I_i_* denotes the importance of the its node.

Eleven network topology parameters (betweenness, bottle neck, closeness, degree, DMNC, eccentricity, EPC, MCC, MNC, radiality, etc.) were chosen based on the features of the network topology. The core module in each group was determined by analyzing the module that ranked highest, and the genes inside that module were found to be the hub genes.

### 4.2. Verification with Hub Genes

An independent GEO dataset was utilized to assess the expression levels of hub genes. Furthermore, the scRNA-Seq data underwent quality control and data filtering. The functions NormalizeData, ScaleData, and RunPCA were used for the purpose of normalizing and performing PCA analysis on the data [37]. The ideal number of principal components (PCs) for the upcoming t-distributed stochastic neighbor embedding (t-SNE) analysis was determined based on the findings of the ElbowPlot and JackStraw analyses. The Cellmarker database (http://xteam.xbio.top/CellMarker/ (accessed on 20 April 2024)) was utilized to obtain the variations between cell subtype genes via the FindAllMarkers method.

### 4.3. Functional Enrichment Analysis

Subsequently, gene ontology (GO) function analysis and Kyoto Encyclopedia of Genes and Genomes (KEGG) pathway enrichment analysis were performed for hub genes within the module after minimum entropy optimization. 

### 4.4. Immune Infiltration Analysis

The CIBERSORT algorithm was used to quantify and assess the immune and stromal cell quantities in the GA and control groups [38]. The Spearman was then used to research the correlation between the cells and the Hub-GA. A single-sample gene set enrichment analysis (ssGSEA) score was calculated for immune cell infiltration and immune-related functions in the two groups.

### 4.5. Effective Drug Combination Screening

The TCMSP database was utilized to conduct a search for the active ingredients of TCM, including 499 herbs, 13,144 compounds, and 785 targets. The compounds were screened, normalized, and deduplicated based on their oral bioavailability (OB) and drug-like characteristics (DL). The thresholds for screening were set at OB ≥ 30% and DL ≥ 0.18.

#### 4.5.1. The Relationship between Chinese Herbal Medicines and Disease under Effectiveness and Overlap

The association between Chinese herbal medicines and hub genes was initially evaluated using the overlap rate [39]. Furthermore, Hscore quantified the efficacy of Chinese herbal medicines in combating diseases [40]. The system established connections across drug combination compounds–disease networks using the PageRank algorithm and a two-step random walk algorithm. The number of components found in TCM and its effect on the target determined the importance of these targets in the PPI network. The effectiveness of Chinese herbal medicines in treating diseases was directly proportional to the number of targets it acts upon and the higher the Hscore. Calculate according to the following formula:(2)$$Overlap\;rate=\frac{\left(A\cap B\right)}{\left(A+B-A\cap B\right)}.$$

The gene numbers A, B, and A∩B reflect the gene numbers of the Chinese herbal medicines, Hub-GA, and the intersection between drug and disease genes, respectively.
(3)$${Hscore\gamma }_{i}=\sum_{\beta =1}^{n}\frac{A_{\beta }V_{\beta \gamma }}{\beta^{out}}.$$

*A_β_* indicates the target score for *β*, and *V_βγ_* signifies the matrix created between *β* and *γ*, where 1 and 0 indicate the presence or absence of a connection between the two. The output value of node *β* in the network has been designated by *β^out^*.

#### 4.5.2. The Relationship between Chinese Herbal Medicines and Disease under the Network Module

Within the protein interaction set, the genes related to each TCM form modules, and the similarity between these modules is shown by their network distance from the disease module [41]. The mean shortest distance (*d_AB_*) between drug–disease target pairings and the mean shortest distance (*S_AB_*) within each group are evaluated in a comparison of network interactions between drug and disease targets.
(4)$$S_{AB}=\langle d_{AB}\rangle -\frac{\langle d_{AA}\rangle +\langle d_{BB}\rangle }{2}.$$

The average shortest path between nodes A and B inside the interaction area is represented by *d_AA_* and *d_BB_*. The two target topologies overlap when *S_AB_* < 0. The two sets of targets were topologically separated when *S_AB_* ≥ 0.

The *z* was marked as a dependable metric for gauging the network proximity between Chinese herbal medicine (*X*) and hub genes (*Y*). The calculation of the shortest path length, denoted as *d*(*x*,*y*), between each drug target (*x*) and the disease target (*y*) was determined by the following formula:(5)$$d\left(X,Y\right)=\frac{1}{\left\|Y\right\|}\sum_{y\in Y}{min}_{x\in X}d\left(x,y\right),$$
(6)$$z=\frac{d-\mu }{\sigma }.$$

The mean was *μ* and the standard deviation was *σ*. Herbs and disease genes separated from each other with a *z* ≥ 0 from a network perspective. Otherwise, *z* < 0.

#### 4.5.3. The Relationship between Chinese Herbal Medicines and Disease under Clinical Medication Experience

Pscore is a measure that indicates the effectiveness of herbal combinations in the use of TCM, based on the clinical experience and habits of experts [40]. The Pscore values of 28,279 TCM pairs were computed by screening, deduplicating, and normalizing the database. Mscore was created to quantify the average frequency of specific Chinese herbal medicines in regularly used combinations.
(7)$$Pscore=count({Herb}_{i}\cap {Herb}_{j}),$$
(8)$$Mscore=\frac{\sum_{n=0}^{\infty }\left({Pscore}_{j}\right)}{n_{j}}.$$

*Herb_i_* and *Herb_j_* refer to TCM, *Pscore_j_* is the sum of all drug pairs containing *Herb_j_* in the database, and *n* is the number of drug pairs found.

#### 4.5.4. Molecular Docking

Molecular docking was performed between compounds that appeared more than five times and hub genes after the chemical components of TCM were gathered utilizing the screening approach mentioned above. The AutoDock Vina 1.1.2 program was utilized to perform docking of the receptor protein and the ligand separately. The affinity was evaluated using molecular docking, which is a crucial sign of a ligand’s ability to attach to a receptor molecule efficiently. 

#### 4.5.5. Identification of Representative Compounds and TCM Combination Refinement with Similarity Clustering

The molecular fingerprints of the active compounds were downloaded. The similarity of molecular fingerprints was evaluated with the Tanimoto coefficient. The compounds in drug combinations were then classified into three groups based on their distance utilizing the K-means clustering. This algorithm has been widely used in the field of drug discovery [42,43]. A disease–TCM compounds–targets–pathways network was constructed, and the Dunn index and silhouette score utilized as evaluation indexes to validate the clustering results. 

#### 4.5.6. Cooperative Association of Herbal Combination Based on SVD

In statistics and machine learning, SVD serves as a common linear transformation method that is frequently used [44]. The four properties and five tastes, meridians, association between diseases and Chinese herbal medicine, and representative compounds were included in the calculation through the construction of the core TCM matrix. The presence of these components was set to 1, and the absence of them to 0. Following the normalization of the aforementioned data, the matrix underwent SVD to obtain the relevant principal components, and the vector inner product was used to compute the correlation coefficient among herbal combinations. This approach creatively took into account TCM’s inherent capacity to regulate disease and compound nodes [45]. Conversely, we devised the following formula to determine the synergy index (*S_i_*) by combining the compounds’ docking ability with the average effective coverage score (*E_c_*) of TCM compounds:(9)$$E_{C}=\frac{1}{N_{Herb}}\cdot \frac{n_{Herb(j^{Rc})}}{n_{Herb(j)}},$$
(10)$$Si=\frac{\left|\sum_{1}^{n_{Herb(j^{Rc})}}A_{j}\right|\cdot d_{j}\cdot E_{c}}{\left|\sum_{1}^{N}s_{ab(j)}\right|\cdot {D_{j}\cdot n}_{j}}.$$

*n* represents the numerical value of the core herbal medicine, *n_Herb_* the number of representative compounds discovered through molecular docking, *n_(Herb(j))_* the number of compounds in *Herb_j_*, and *n_(Herb(j_^Rc^_))_* the number of representative compounds in *Herb_j_*. *A_j_* represents TCM representative compound molecules’ binding energy, *d_j_* the shortest distance between TCM and disease interaction, *s_(ab(j))_* the network distance between Chinese herbal medicines, *D_j_* the maximum clustering distance, and *n_j_* the number of TCM targets.

### 4.6. Method Comparison

To evaluate the model’s performance, we compared it with five common machine learning models, RF [46], GBDT [47], SVM [48], XGBoost [49], and CART [50], using a 5-fold cross-validation (CV) strategy. Since these models rely on matrix-like feature vectors, the 5-fold CV method was directly applied to assess their performance. The primary outcome measure used was the MSE. In addition, we report the root-mean-square error (RMSE), mean absolute error (MAE), and MAPE to compare the predicted values with the true values. The data were split with a training-to-test set ratio of 8:2.

### 4.7. Drug Combinations Analysis Utilizing UPLC-Q-TOF-MS/MS

Effective drug combinations treating GA (GAD) used the following: Tamaricis Cacumen, Cuscutae Semen, Artemisiae Annuae Herba, Schizonepetae Herba, Gleditsiae Spina, Prunellae Spica, and Achyranthis Bidentatae Radix. The Chinese National Pharmacopoeia’s standards were fulfilled, and it was procured from a local source. To make a GAD water decoction, soak the herbs for 30 min beforehand, then add nine times the volume of water and boil for 2 h. After filtering the decoction and continuing to boil it to obtain a raw drug concentration of 1 g/mL, the mixture was once more filtered, sterilized, and stored at 4 °C for subsequent experimentation.

For detection and analysis by mass spectrometry, the 500 μL GAD was filtered through a 0.22 μM membrane. The determination was performed on an ACQUITY UPLC-BEH C18 column (2.1 mm × 100 mm, 1.7 μm) using the Waters ACQUITY UPLCTM system. The mobile phase consisted of 0.1% acetonitrile solution (B) and 0.2% formic acid aqueous solution (A). The sample size was 1 μL, and the flow rate was 300 μL/min. Mobile phase A consisted of 0.2% formic acid aqueous solution (A)–acetonitrile (B); gradient elution took place for 0–30 min for 5–95% B and 30–33 min for 95% B–5% B. Principal component analysis scanning mode was MS^e^, the electrospray ion source was employed, and negative ion mode was used for detection. The ion source’s operational parameters were as follows: The spectral acquisition interval was 0.2 s, the scanning range was m/z 50–1500, the capillary voltage was 3.0 kV, the cone hole voltage was 40 V, the solvent gas flow was 800 L·h^−1^, the ion source temperature was 120 °C, and the solvent removal temperature was 450 °C. The cone hole backblowing gas flow (N_2_) was 50 L·h^−1^.

### 4.8. Verification of Pharmacodynamic Experiment

#### 4.8.1. Animals and Drugs

Forty male C57BL/6J mice, aged 6–8 weeks, were used in this experiment. The mice were bred in a sterile animal breeding facility at Zhongyan Zichuang (Beijing) Biotechnology Co., Ltd., Beijing, China. The research was conducted in accordance with the standards set by the Ethics Committee, with the approval number ZYZC20240613S.

Positive medication, indomethacin enteric-coated pills, manufactured by Linfen Baozhu Pharmaceutical Co., Ltd., ShanXi, China; lot number: 200301. Specification per tablet: 25 mg. For subsequent usage, make a 0.3 g/mL liquid with regular saline and store it in the refrigerator at 4 °C.

#### 4.8.2. Animal Modeling, Grouping, and Drug Administration

Mice were provided with food and water to ensure their survival, implementing adaptive feeding for one week before the formal experiment. Preparation of MSU crystals: following the Coderre recording method, a suspension of MSU crystals with equal concentration was manufactured [51]. The model was built by injecting the suspension vertically using a needle along the lateral rear of the ankle joint of mice. Following the random number table method, eight mice were divided into each of the following groups: the control (Con), model (Mod), cirsiliol (Cir), indomethacin (In), and GAD. The concentration of the substance used in GAD was 3 g/kg/d after a 7-day continuous intervention, while the concentration of the In that was found to be effective was 0.3 g/mL/d. The Cir dosage was 25 mg/mL/d with a 10 mL/kg dosing volume.

#### 4.8.3. Ankle Joint Swelling Index

The circumference of the ankle joint in each group of mice was measured using vernier calipers, and the ankle joint swelling index was computed. Ankle joint swelling index = joint thickness at each point in time after administration − thickness before administration (mm).

#### 4.8.4. Hematoxylin and Eosin (HE) Staining

The tissues in each group were fixed with 4% paraformaldehyde (Solarbio, Beijing, China), decalcified, dehydrated, waxed, embedded, sliced, and stained with HE (Beyotime, Shanghai, China) in order to assess alterations in the morphology of the kidneys and ankle joints.

#### 4.8.5. Enzyme-Linked Immunosorbent Assay (ELISA)

Each mice group’s serum was obtained, and the levels of uric acid (UA) and associated inflammatory components were measured. Interleukin-1α (IL1A) and interleukin-6 (IL6) content and UA were measured using IL1A, IL6, and UA kits (Proteintech, Wuhan, China, Nanjing Jiancheng, Nanjing, China).

#### 4.8.6. RNA Extraction and Quantitative Polymerase Chain Reaction (q-PCR)

Using column extraction, each animal group’s RNA was extracted in compliance with the RNA extraction kit’s instructions (Vazyme, Nanjing, China). Each group of RNA samples was examined using a microspectrophotometer to ascertain the purity and concentration. The PCR amplification system and reverse transcription system were configured in compliance with the instructions provided by the appropriate kit’s manufacturer (Vazyme, China). The temperatures utilized in the PCR reaction were pre-denaturation at 95 °C for 30 s, cycle reaction at 95 °C for 10 s, and 60 °C for 30 s. The entire process was calibrated using GAPDH. The mRNA expression levels of the important target proteins TGFB1, PTGS2, and MMP3 were measured using the 2^−ΔΔCt^ method. Shanghai Jierui Bioengineering Co., Ltd. (Shanghai, China) supplied the primers. Table 3 shows a list of the primer sequences used.

#### 4.8.7. Western Blotting

Each animal group’s ankle joints were given a RIPA lysate containing PMSF, which was then centrifuged, the supernatant was aspirated, and 5× protein-loading buffer was added for denaturation. A 6–15% sodium dodecyl–sulfate polyacrylamide gel electrophoresis (SDS-PAGE) gel was used for protein electrophoresis. The proteins were transferred onto 0.45 μm PVDF membranes using wet transfer and blocked with 5% skimmed milk powder. The antibodies were diluted according to the manufacturer’s instructions, and the membranes were incubated with TGFB1 (1:2000; Proteintech, China), PTGS2 (1:1000; Proteintech, China), and MMP3 (1:1000; Proteintech, China) as the primary antibodies and GAPDH antibody (1:5000; Proteintech, China) as an internal control. The membrane was then incubated with the corresponding secondary antibody (1:2000; Proteintech, China). ECL reagent (Beyotime, China) was added for development, and visualization was performed using a chemiluminescence imaging system. The ImageJ was used to analyze the optical densities of the target bands.

### 4.9. Data Sources and Statistical Methods

Disease databases related to GA and gout gene expression were downloaded from the GEO database (https://www.ncbi.nlm.nih.gov/geo/ (accessed on 20 April 2024)). The keywords “gouty arthritis” and “gout” were used to search for the following datasets: GSE199950, GSE242872, GSE217561, and GSE160170. The RNA-seq data were extracted using the fragment per kilo base million (FPKM) format, and the log2 (FPKM + 1) transform was used for standardizing the data. The drug datasets were obtained from the TCMSP database (https://tcmsp.91medicine.cn/ (accessed on 20 April 2024)) and Drugbank database (https://go.drugbank.com/ (accessed on 20 April 2024)). The UniProt database (http://www.uniprot.org/ (accessed on 20 April 2024)) was utilized to standardize and eliminate duplicate entries of the acquired effective targets. Clinical application experience data for TCM was taken from the Chinese patent medicine database (http://crds.release.daodikeji.com (accessed on 20 April 2024)). The chemical composition was determined using the 3D structure in the PubChem database (https://pubchem.ncbi.nlm.nih.gov/ (accessed on 20 April 2024)), while the equivalent structure of receptors was found in the Protein Data Bank database (http://www.rcsb.org/pdb (accessed on 20 April 2024)).

Software, including Python 3.11 and GraphPad Prism 8.0.1, was used to conduct statistical analysis. The mean ± standard error of the mean (SEM) was used to express all measurement data. Comparisons between the two groups were performed using independent sample *t*-tests. Nonparametric tests were used for samples that did not conform to normal distribution. Statistical significance was set at *p* < 0.05.

## 5. Conclusions

In summary, we have developed a novel multi-level network framework named Syn-COM for screening potential drug combinations for disease treatment. By employing various bioinformatics algorithms and module partitioning, this framework filters and identifies key hub genes related to diseases, offering insights into the diversity and complexity of immune and biological functions. The drug-screening process integrates network features, overlap degree, molecular docking, and similarity to efficiently extract data features and identify optimal drug combinations, independent of dose relationships. The framework also innovatively calculates drug interaction and association using similarity clustering and synergy matrix filtering. This method is versatile and applicable to predicting drug combinations beyond Western medicine for various diseases. By incorporating clinical experience and TCM characteristics, the framework enhances the prediction of TCM combinations. Using GA as a case study, preliminary experimental validation suggests that the herbal combination GAD could be significant in future GA treatment. This approach shows great potential for identifying drug combinations and developing innovative therapies for a wide range of human diseases.

## Figures and Tables

**Figure 1 pharmaceuticals-17-01230-f001:**
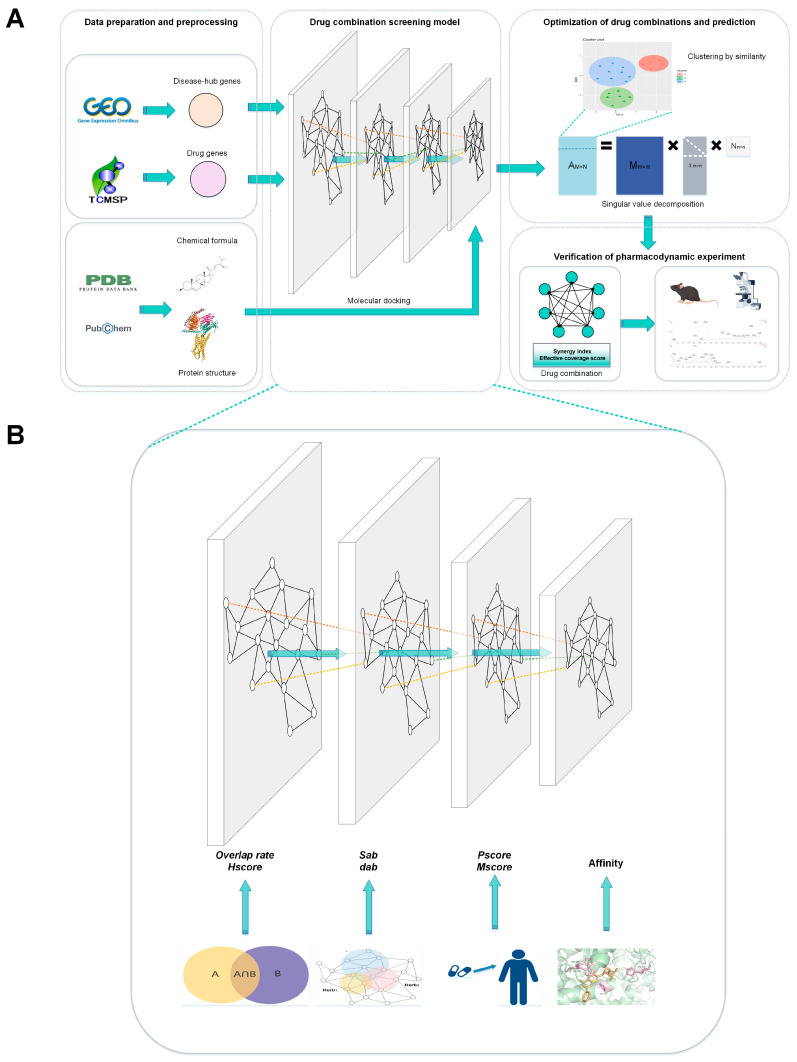
(**A**) The whole research’s experimental technical road map. (**B**) Detailed methodology for the screening model of drug combinations.

**Figure 2 pharmaceuticals-17-01230-f002:**
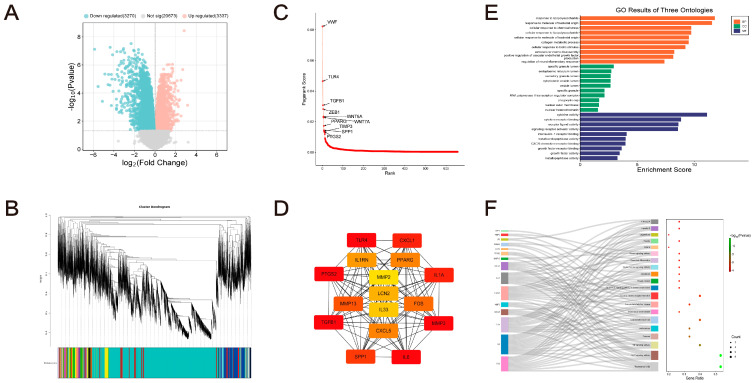
The hub genes related to GA (Hub-GA)’s screening and functional analysis. (**A**) Volcano plot of DEGs in GSE199950. (**B**) The clustering tree was analyzed with WGCNA, with various colors signifying various modules. (**C**) Utilizing random walks to sequence advantage genes. (**D**) The MCC score escalates as the color transitions to red, with elevated node scores signifying greater significance within the network. (**E**,**F**) Analysis of GO and KEGG.

**Figure 3 pharmaceuticals-17-01230-f003:**
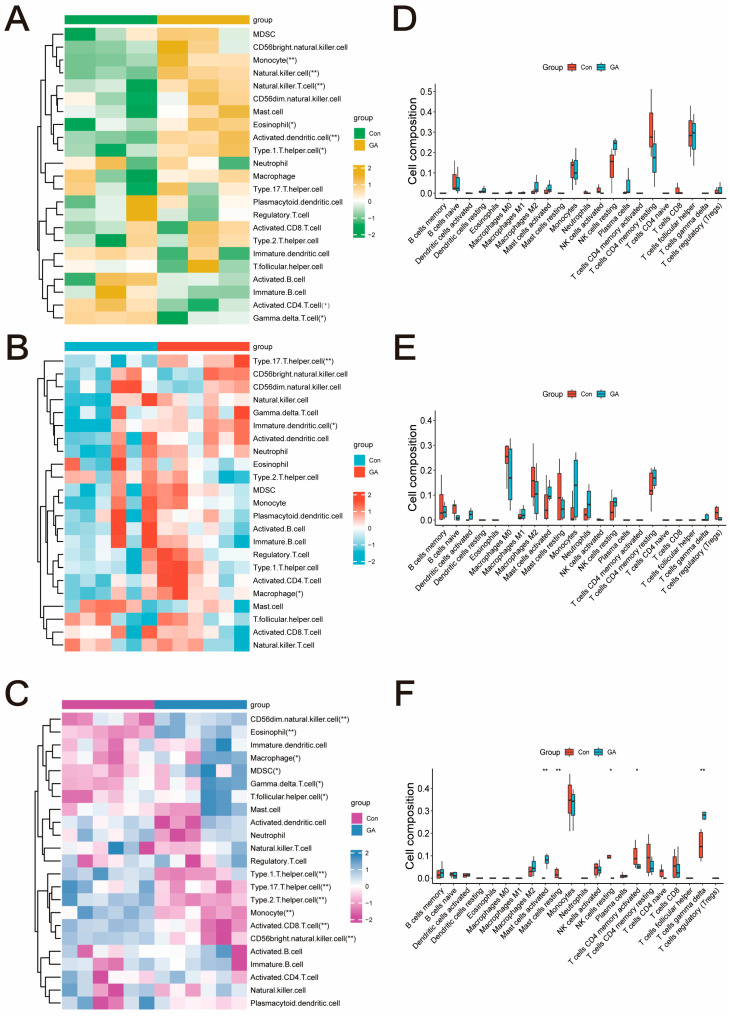
Immune infiltration analysis. (**A**,**D**) GSE199950. (**B**,**E**) GSE242872. (**C**,**F**) GSE160170. * *p* < 0.05 vs. Con; ** *p* < 0.01 vs. Con.

**Figure 4 pharmaceuticals-17-01230-f004:**
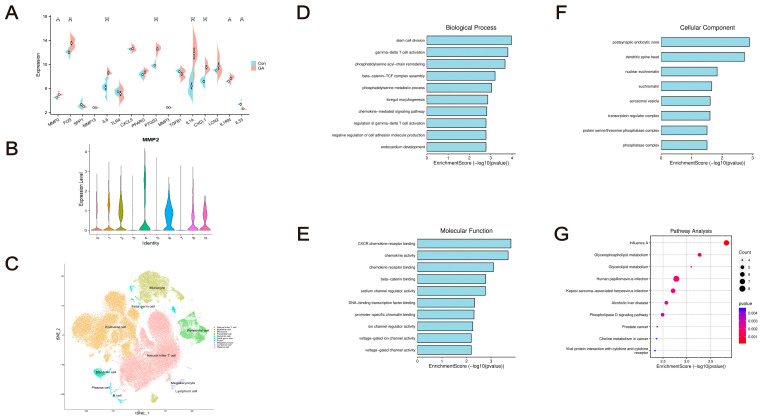
Hub-GA data set validation. (**A**) Based on the GSE160170 data set, variations in Hub-GA expression between the normal group and the GA group. (**B**) MMP2 expression level in each cell. (**C**) t-SNE map analysis for each cell. (**D**) BP. (**E**) CC. (**F**) MF. (**G**) KEGG analysis. * *p* < 0.05 vs. Con; ** *p* < 0.01 vs. Con; *** *p* < 0.001 vs. Con.

**Figure 5 pharmaceuticals-17-01230-f005:**
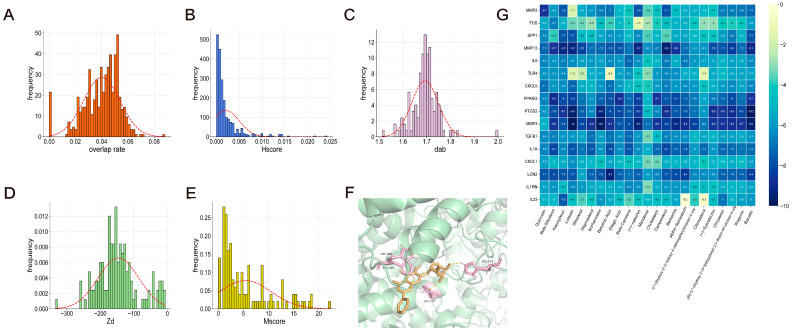
The pertinent Chinese herbal medicines screening criteria for GA intervention under several dimensions: (**A**) Overlap rate. (**B**) Hscore. (**C**) d_ab_. (**D**) Z_d_. (**E**) Mscore. (**F**) Molecular docking diagram of PTGS2 and Baicalin. (**G**) Heatmap of molecular docking.

**Figure 6 pharmaceuticals-17-01230-f006:**
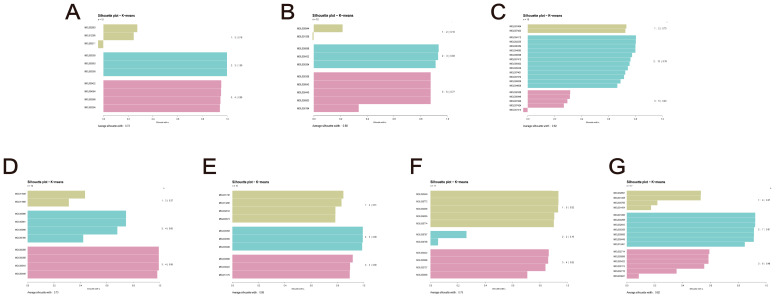
Similarity clustering-based collaborative TCM filtering. (**A**) Tamaricis Cacumen. (**B**) Cuscutae Semen. (**C**) Artemisiae Annuae Herba. (**D**) Schizonepetae Herba. (**E**) Gleditsiae Spina. (**F**) Prunellae Spica. (**G**) Achyranthis Bidentatae Radix.

**Figure 7 pharmaceuticals-17-01230-f007:**
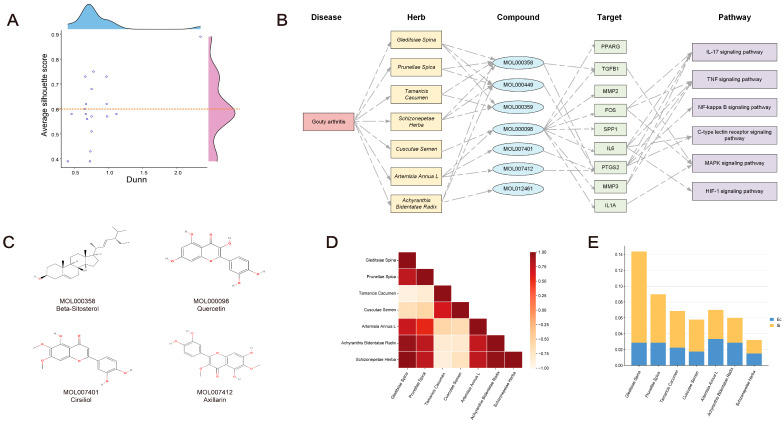
Building a network of disease–TCM compound–target signaling pathways through collaborative association. The following examples show the relationship between the Dunn and average silhouette scores of Chinese herbal medicines: (**A**) Scatter plot. (**B**) Building multi-layer networks. (**C**) High-frequency compounds and recently identified targeted compounds. (**D**) Heatmap of each Chinese herbal medicine’s correlation coefficient. (**E**) Histogram of each Chinese herbal medicine’s Ec and S_i_ in GAD.

**Figure 8 pharmaceuticals-17-01230-f008:**
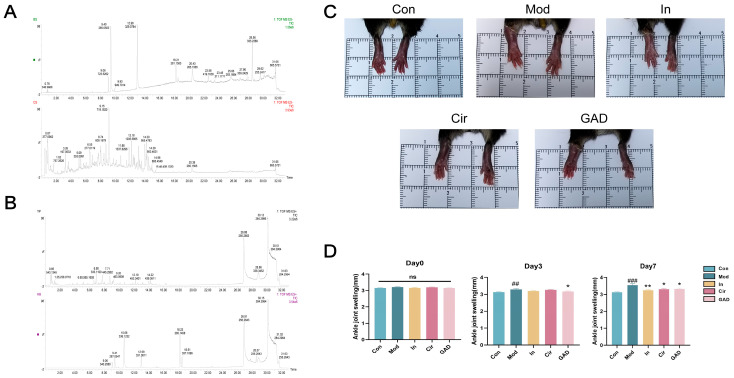
Depiction of GAD components and variations in ankle joint swelling in mice previous to and following GA administration. (**A**,**B**) TIC of GAD by UPLC-Q-TOF-MS/MS. (**C**) Mice’s ankle joint swelling after seven days of treatment in each group. (**D**) Each group’s ankle joint swelling index at each time point. ## *p* < 0.01 vs. Con; ### *p* < 0.001 vs. Con; * *p* < 0.05 vs. Mod; ** *p* < 0.01 vs. Mod.

**Figure 9 pharmaceuticals-17-01230-f009:**
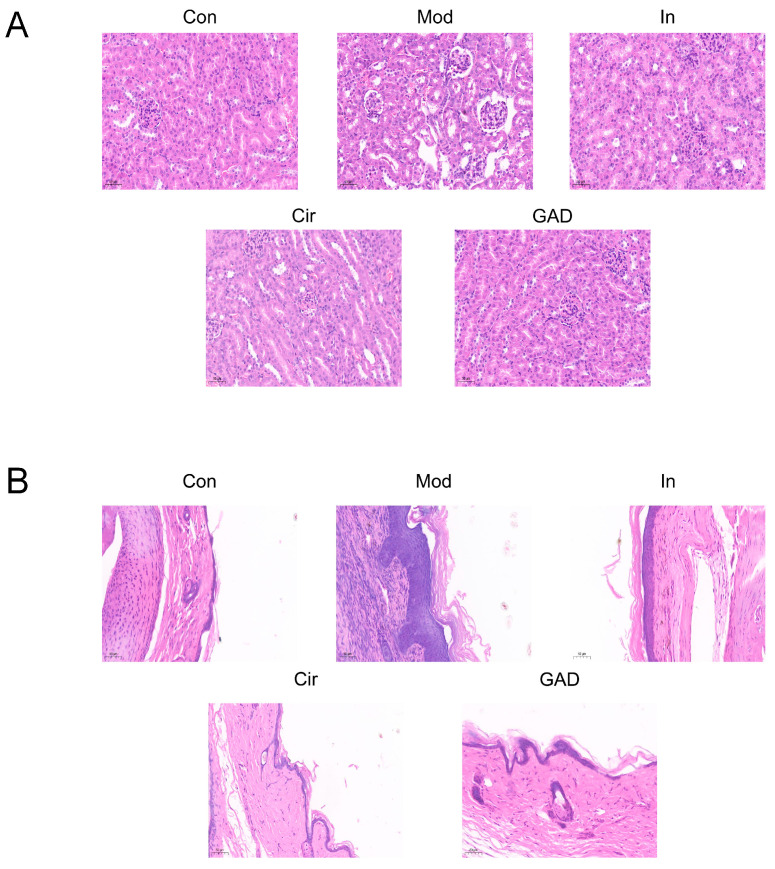
Modifications in kidney (**A**) and joint tissue (**B**) morphology in each mouse group (×200).

**Figure 10 pharmaceuticals-17-01230-f010:**
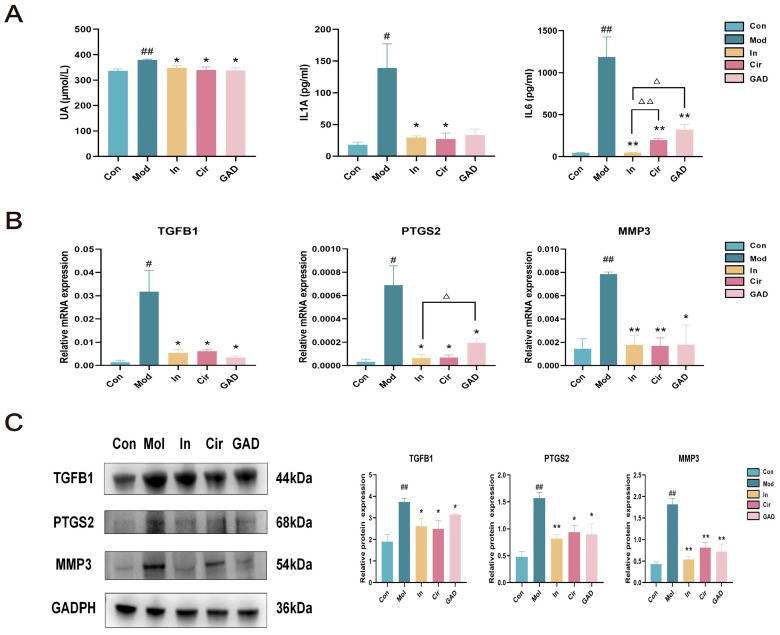
The effect of GAD modulation on aberrant inflammation in GA mice. (**A**) ELISA was utilized to identify the levels of UA and inflammatory-related indices for each group. (**B**) q-PCR was utilized to determine the expression levels of TGFB1, PTGS2, and MMP3 in each group. (**C**) GAD inhibited excessive inflammation by modulating TGFB1, PTGS2, and MMP3 expression. # *p* < 0.05 vs. Con; ## *p* < 0.01 vs. Con; * *p* < 0.05 vs. Mod; ** *p* < 0.01 vs. Mod; Δ *p* < 0.05 vs. In; ΔΔ *p* < 0.01 vs. In.

**Table 1 pharmaceuticals-17-01230-t001:** Information of a potential TCM combination to intervene in GA.

Name	Overlap Rate	Hscore	d_ab_	Z_d_	Mscore	Silhouette Score
*Gleditsiae Spina*	0.0462	0.0023	1.7059	−175.1157	3.0152	0.89
*Prunellae Spica*	0.0464	0.0016	1.7094	−137.5972	4.4550	0.75
*Tamaricis Cacumen*	0.0484	0.0014	1.6821	−177.3917	2.2708	0.73
*Schizonepetae Herba*	0.0474	0.0012	1.7186	−126.7836	8.7373	0.73
*Cuscutae Semen*	0.0448	0.0028	1.7143	−173.9090	14.7876	0.68
*Achyranthis Bidentatae Radix*	0.0455	0.0058	1.7053	−150.4610	15.6081	0.62
*Artemisiae Annuae Herba*	0.0437	0.0057	1.7163	−167.2389	3.3503	0.62

**Table 2 pharmaceuticals-17-01230-t002:** Results of method comparison on the regression task.

Name	MSE	RMSE	MAE	MAPE
RF	0.007	0.063	0.056	8.570
GBDT	0.010	0.101	0.088	13.787
SVM	0.009	0.096	0.077	13.687
XGBoost	0.010	0.099	0.090	14.086
CART	0.000	0.011	0.010	35.845
Syn-COM	**0.001**	**0.037**	**0.025**	**0.981**

**Table 3 pharmaceuticals-17-01230-t003:** Primers sequences used for quantitative PCR.

Gene	Primers
TGFB1	Forward-5′-TGATACGCCTGAGTGGCTGTCT-3′
	Reverse-5′-CACAAGAGCAGTGAGCGCTGAA-3′
PTGS2	Forward-5′-GATCCCCAGGGCTCAAACAT-3′
	Reverse-5′-GAAAAGGCGCAGTTTACGCT-3′
MMP3	Forward-5′-CACTCACAGACCTGACTCGGTT-3′
	Reverse-5′-AAGCAGGATCACAGTTGGCTGG-3′
GAPDH	Forward-5′-TCTTGCTCAGTGTCCTTGC-3′
	Reverse-5′-CTTTGTCAAGCTCATTTCCTGG-3′

## Data Availability

Publicly available datasets were analyzed in this study. These data can be found here: https://www.ncbi.nlm.nih.gov/ (accessed on approval 18 April 2024); GSE199950, GSE242872, GSE217561, GSE160170.

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
