# Peer review of "Syn-COM: A Multi-Level Predictive Synergy Framework for Innovative Drug Combinations"

_pharmaceuticals, 2024, doi:10.3390/ph17091230_

Round 1

Reviewer 1 Report

Comments and Suggestions for Authors

Entitled manuscript " Constructing a multi-level predictive synergy framework for 2 innovative drug combinations: via gouty arthritis as a model" presents the application of bioinformatics methods and large-scale modeling methods for drug prediction and disease treatment.

It is imperative for the authors to thoroughly revise the manuscript to address the below given comments before publication.

1. Write a full-fledged abstract, remove the blocks materials, results, conclusions from the initial field. Show the problems of this article and what features are used in forecasting.

2. Page n. 2, “We aim to provide solid and trustworthy methods for the TCM treatment of GA (Figure1)” - As far as can be seen, the authors are trying to illustrate the screening scheme, or rather the main stages. It is very difficult to perceive, and most importantly, a significant revision is needed. I recommend dividing it into several stages, taking into account the clarity of the image.

3. Regarding the rest of the images. It is very difficult to make out the small text and identify the patterns described by the authors in Figures 2-8. Please fix it.

4. “Utilizing K-means clustering and stratification, additional research of Chinese herbal medicines containing more than five active compounds revealed that the average silhouette score and dunn mean were 0.5989 and 0.8398” - How legitimate is it to compare Silhouette score by one such indicator? What explains the choice as the main unit? There are other characteristics of the selected ones. Use a comparison on them too.

5. How important is the use of these gene primers. How it depends on finding others. What is the peculiarity of this extraction?

6. Are the databases that are used in the public domain? How accessible and reliable are these sources? (https://www.ncbi.nlm.nih.gov/geo/). (http://lsp.nwu.edu.cn/tcmsp.php) (https://go.drugbank.com/). (http://www.uniprot.org/) lease.daodikeji.com).

7. The conclusions omitted the main purpose of this study. It is necessary to study the features of docking more globally and in full detail. Present it in an expanded form without unnecessary details.

8. How does the structure-properties setting for connections depend? I would like to see a more structured text and applicability to well-known drugs, comparison with them, perhaps with other works in this field. The authors declare the treatment of diseases. I hope these are not only words, but also evidence-based results.

Author Response

Comments 1: Write a full-fledged abstract, remove the blocks materials, results, conclusions from the initial field. Show the problems of this article and what features are used in forecasting.

Response 1: Thank you for your reminder. These comments are all of great importance to our article. We revised the abstract by omitting sections on background, results, and conclusions from the initial draft, instead emphasizing the framework's functionality, methods, and validation results using gouty arthritis as an example. This revision highlights the framework's applicability and accuracy. It can be found on lines 13-29 of the Page 1.

Comments 2: Page n. 2, “We aim to provide solid and trustworthy methods for the TCM treatment of GA (Figure1)” - As far as can be seen, the authors are trying to illustrate the screening scheme, or rather the main stages. It is very difficult to perceive, and most importantly, a significant revision is needed. I recommend dividing it into several stages, taking into account the clarity of the image.

Response 2: We feel great thanks for your professional review work on our article. We have made substantial revisions to the flowchart in Figure 1, now divided into two sections. The updated flowchart emphasizes the core process of constructing the drug combination screening framework, showcasing the innovative use of various methods for extracting specific data indicators. Additionally, we ensured the image's clarity by uploading high-resolution versions in both JPG and TIFF formats in the supplementary materials.

Comments 3: Regarding the rest of the images. It is very difficult to make out the small text and identify the patterns described by the authors in Figures 2-8. Please fix it.

Response 3: Thank you for the reviewer's suggestions. To enhance the visibility of the text in the images within the manuscript, we have re-uploaded the images. Furthermore, each image in the consolidated graph is meticulously summarized and structured into a document of Figure Legends, enhancing clarity and convenience for reviewers during their consultation. Additionally, for the convenience of reviewers and editors during the subsequent review process, we have included high-resolution versions of each image in JPG format in the Supplementary Materials.

Comments 4: “Utilizing K-means clustering and stratification, additional research of Chinese herbal medicines containing more than five active compounds revealed that the average silhouette score and dunn mean were 0.5989 and 0.8398” - How legitimate is it to compare Silhouette score by one such indicator? What explains the choice as the main unit? There are other characteristics of the selected ones. Use a comparison on them too.

Response 4: Thank you for your nice comments on our article. The average silhouette score and Dunn index are numerical evaluation metrics included in the K-means and rcdk packages, as cited in the original references within the Materials and Methods section. The Silhouette coefficient, a clustering validation method, combines both cohesion and separation. It measures the average distance of each point to other points within its cluster and to points in neighboring clusters. Silhouette values range from [-1,1], where a value close to 1 indicates that the point is well separated from neighboring clusters and closely aligned with its assigned cluster. Higher values are preferable. The Dunn index is the ratio of the smallest inter-cluster distance to the largest intra-cluster distance, ranging from 0 to infinity, with higher values indicating compact and well-separated clusters.

As shown in our analysis, we fitted each index separately, with the Silhouette coefficient outperforming the Dunn index. The Silhouette coefficient was selected as it evaluates both intra-cluster cohesion and inter-cluster separation, offering a more comprehensive evaluation of the dataset's clustering. The peak value of the average silhouette score was 0.5989, and we chose scores greater than 0.6 as the threshold for subsequent screening. To better emphasize the screening process based on this coefficient, we optimized Figure 7A and revised the descriptions in the Materials and Methods and Results (Page 7, Lines 179-187; Page 19, Lines 501-506) sections. The raw data and images have been submitted in the Supplementary Materials as Figures S1-2.

Comments 5: How important is the use of these gene primers. How it depends on finding others. What is the peculiarity of this extraction.

Response 5: Thank you for the reviewer's suggestions. The selection of gene primers in this study was based on the following reasons:

1) As depicted in Figure 2D, TGFB1, PTGS2, and MMP3 were identified as key hub genes in the gene screen for gouty arthritis (GA). The MCC algorithm revealed that these genes exhibit high centrality in the protein interaction network, with darker colors indicating a greater degree of centrality and higher node importance (Page 4, Lines 108-110, 115-117). Furthermore, IL6 and IL1A were detected in serum, and their expression levels were measured using ELISA to validate the accuracy of the hub gene screening.

2) As shown in Figure 4A, validation of the hub genes indicated a significant correlation between the expression differences of PTGS2, IL6, and IL1A in the gouty arthritis group and the control group (P<0.05; Page 5, Lines 134-135). Additionally, molecular docking results in Figure 4G demonstrated strong docking abilities for PTGS2 and MMP3 with effective compounds (Page 7, Lines 176-177).

3) PTGS2, also known as Cyclooxygenase-2 (COX-2), is involved in the conversion of arachidonic acid to prostaglandin H2 and plays a role in various physiological and pathological processes. It is a well-known target of non-steroidal anti-inflammatory drugs (NSAIDs) and is considered crucial for the treatment of gouty arthritis. In this study, cirsiliol and axillarin were found to target PTGS2, highlighting the importance of validating the PTGS2 gene (Page 7-8, Lines 199-203).

4) As illustrated in Figure 7B, the compounds MOL000358, MOL000449, and MOL000098 were highly present in the TCM combination GAD, as confirmed by mass spectrometry analysis. These potent compounds target TGFB1 and PTGS2. Therefore, based on this evidence, we selected the gene primers and proteins to validate the expression levels of PTGS2, TGFB1, and MMP3 in animal models.

To further clarify the rationale behind our gene selection and underscore the importance of these genes, we have added supplementary results to the discussion section (Page 10, Lines 263-265).

Comments 6: Are the databases that are used in the public domain? How accessible and reliable are these sources? (https://www.ncbi.nlm.nih.gov/geo/). (http://lsp.nwu.edu.cn/tcmsp.php) (https://go.drugbank.com/). (http://www.uniprot.org/) lease.daodikeji.com).

Response 6: We would like to express our gratitude to the reviewers for their valuable suggestions. In response to point 6, we have made the following revisions. The databases referenced in this study, including GEO, TCMSP, DrugBank, UniProt, PubChem, PDB, and others, are publicly accessible and widely used in life science research. These databases are considered reliable sources of information for academic and industrial applications.

The GEO database contains data submitted by researchers and institutions worldwide, often associated with published studies, ensuring high data quality. GEO2R, a data processing tool within the GEO platform, allows for further analysis. DrugBank provides comprehensive information on drugs and drug targets, while the TCMSP integrates literature and experimental studies to offer insights into traditional Chinese medicine (TCM) components, targets, and pharmacokinetics. UniProt is a highly authoritative resource for protein sequences and functional data, featuring standardized formats and detailed annotations. PubChem compiles experimental data, literature, and computational information on chemical molecules and biological activities. The PDB database, which primarily includes experimentally determined structures (e.g., X-ray crystallography, NMR), is crucial for structural biology research due to its high reliability. The Chinese Patent Medicine Value Assessment Information Database is a specialized tool for evaluating Chinese patent medicine, providing data on components, efficacy, safety, quality, and market value, thus enhancing its clinical utility.

These databases are widely regarded as authoritative in their respective fields. To ensure the accuracy and reliability of the data, we implemented multiple screening thresholds, thereby maintaining the integrity of the results.:

Comments 7: The conclusions omitted the main purpose of this study. It is necessary to study the features of docking more globally and in full detail. Present it in an expanded form without unnecessary details.

Response 7: We sincerely appreciate your valuable suggestions. In response, we have revised the conclusion section to present it in a more comprehensive and concise manner. The revised conclusion now provides a more detailed summary of the model's characteristics, key indicators, functions, and adaptability, ensuring a clearer and more thorough representation without unnecessary detail. The concluding section is described on Page 19, lines 637-651.

Comments 8: How does the structure-properties setting for connections depend? I would like to see a more structured text and applicability to well-known drugs, comparison with them, perhaps with other works in this field. The authors declare the treatment of diseases. I hope these are not only words, but also evidence-based results.

Response 8: We sincerely thank the reviewers for their valuable suggestions. In response, we clarify that the model proposed in this manuscript does not rely solely on structural relationship-based screening. The molecular docking simulations presented are only one aspect of the filtering process, used to identify associations between the active ingredients of traditional Chinese medicine (TCM) and hub genes. Additionally, this approach incorporates the distance and overlap within the disease-drug network from the previous step, along with the comprehensive normalization of clinical drug use and feature selection results. This combination improves both the model’s efficiency and generalizability. A more detailed explanation of the model’s construction is provided in Figure 1B, Page 8 (Lines 218-227), and Page 16 (Lines 525-532).

Notably, current models for screening TCM combinations based on non-structural relationships are lacking, which represents a key innovation of this study. We also compared the performance of the proposed model with common machine learning models, such as Random Forest (RF), Gradient Boosting Decision Trees (GBDT), Support Vector Machine (SVM), XGBoost, and Classification and Regression Trees (CART), for regression correlation coefficients. Our results show that the proposed model demonstrates superior and more comprehensive performance, supported by extensive research data and rigorous evaluation metrics, making the conclusions highly reliable and scientifically robust.

Response to Comments on the Quality of English Language

We sincerely appreciate the reviewers' feedback regarding the quality of English in our manuscript. In response, we have meticulously reviewed and revised the text to enhance clarity, grammar, and overall readability. We have ensured that the language employed adheres to academic standards, thereby making the manuscript more accessible to a wider audience. Should any specific sections require further refinement, we remain open to undertaking additional revisions to elevate the quality of the language.

Additional clarifications

We appreciate the opportunity to address the reviewers' comments and provide additional clarifications. Here are some details that may help in understanding our revisions:

1. Revised Sections: We have thoroughly revised key sections of the manuscript, including the abstract, introduction, methods, results, and discussion. These revisions aim to enhance the coherence and readability of the text, ensuring that complex concepts are conveyed clearly and concisely.

2. Model Description: We have expanded our explanation of the model construction and its innovative aspects. Specifically, we have clarified how the model integrates molecular docking simulations with distance and overlap measures, as well as clinical drug use normalization, to improve its efficiency and generalizability.

3. Figure and Table Updates: We have updated and optimized all figures and tables to improve visual clarity and ensure alignment with the revised text. Each visual element has been re-evaluated to accurately reflect the data and support the manuscript’s conclusions.

4. Language Quality: To ensure the language meets the high standards required for academic publication, we have engaged a professional editor to polish the manuscript.

5. Additional Data and References: We have included additional references and data as necessary to bolster our claims and provide a more robust foundation for our conclusions.

6. Response to Specific Comments: We have addressed each of the reviewers' specific comments in detail, providing explanations and justifications where required. All changes and responses are documented in the revised manuscript and the accompanying response letter.

We hope these clarifications will assist in the review process and welcome any further feedback to enhance the quality of our work.

Reviewer 2 Report

Comments and Suggestions for Authors

The manuscript titled "Constructing a Multi-Level Predictive Synergy Framework for Innovative Drug Combinations: via Gouty Arthritis as a Model" describes and demonstrates, through an example, how an interdisciplinary approach, supported by artificial intelligence tools, gene analysis, and connection with real medical needs, can open up the field of developing new therapeutic approaches.

The manuscript is very well prepared with many graphical elements. Below, I am sending a few suggestions for improvement.

1. Figure 1 contains a large number of graphical elements, making it difficult to follow the entire process. It is recommended that the illustration be simplified by reducing the number of elements and using clearer, more concise process elements that facilitate easier tracking of the information flow. Consider breaking the figure into multiple panels or using flowcharts to enhance clarity.

2. The manuscript is based on the use of many different packages in the R programming language, with some parts related to Python and others to various other programs. It would be beneficial if the authors could more clearly present the flow of information and systematize the presentation of the computational packages. In its current form, it is very difficult to follow the methodology that was used.

3. It is suggested that the authors include references related to the computational methodology that has been used by other authors in the past.

4. The authors mention an innovative approach in the title, but it would be beneficial to clearly highlight what exactly is innovative, which specific steps are innovative, and how they differ from the current state of the art, rather than just making a general statement about innovation.

5. Overall, the conclusion is written very sparsely. Considering the extensive nature of the paper, the conclusion could be much better formulated.

Author Response

Comments 1: Figure 1 contains a large number of graphical elements, making it difficult to follow the entire process. It is recommended that the illustration be simplified by reducing the number of elements and using clearer, more concise process elements that facilitate easier tracking of the information flow. Consider breaking the figure into multiple panels or using flowcharts to enhance clarity.

Response 1: We sincerely appreciate the reviewers' valuable suggestions. In response, we made significant revisions to Figure 1 to simplify the illustration, reducing the number of elements and using clearer, more concise components to enhance the flow of information. The flowchart is now divided into two steps with an accompanying legend, which not only presents the overall research process but also highlights the primary workflow of the framework described in the manuscript. For ease of reference, all images have been uploaded to the Supplementary Material in JPEG format for convenient access by reviewers and editors.

Comments 2: The manuscript is based on the use of many different packages in the R programming language, with some parts related to Python and others to various other programs. It would be beneficial if the authors could more clearly present the flow of information and systematize the presentation of the computational packages. In its current form, it is very difficult to follow the methodology that was used.

Response 2: We appreciate the reviewers' suggestions regarding the use of various programming packages. Based on their feedback, we have streamlined the code by unifying the Python and R implementations and ultimately presenting all computational aspects in Python for clarity and consistency. A revised version of this section can be found on page 19, lines 631-632. Additionally, we have uploaded the revised code to the Supplementary Material for reference.

Comments 3: It is suggested that the authors include references related to the computational methodology that has been used by other authors in the past.

Response 3: We appreciate the reviewers' valuable suggestions. In response, we have ensured that all references to others' calculation methods mentioned in the Materials and Methods section are appropriately cited. These citations and corresponding revisions can be found on the following pages and lines: Page 13, Lines 390-392 and 406-408; Page 14, Lines 416-418, 429-430, and 440-441; Page 15, Lines 468-470 and 484-485; and Page 16, Lines 508-509 and 515-518.

Comments 4: The authors mention an innovative approach in the title, but it would be beneficial to clearly highlight what exactly is innovative, which specific steps are innovative, and how they differ from the current state of the art, rather than just making a general statement about innovation.

Response 4: We sincerely thank the reviewers once again for their invaluable suggestions. Based on the feedback, we have revised the presentation of the innovative methods described in the manuscript, particularly in the abstract, discussion, and conclusion sections. These revisions emphasize the novelty of our proposed model. The drug screening process now integrates network features, molecular docking, similarity clustering, and overlap analysis to efficiently extract data features and identify optimal drug combinations independently of dose relationships. Notable revisions can be found on Page 1-2, Lines 43-49; Page 2, Lines 79-89; Page 12, Lines 306-307 and 324-339; and Page 13, Lines 375-380.

Additionally, the framework incorporates innovative methods such as similarity clustering and synergy matrix filtering to compute drug interactions and associations, making it adaptable for predicting drug combinations beyond traditional Western drugs. When applied to traditional Chinese medicine (TCM) combinations, the model integrates clinical experience and TCM characteristics, enhancing its predictive power. We have also supplemented the evaluation with a comparison against other common machine learning models, demonstrating that our framework outperforms these models in terms of comprehensive performance. These updates can be found on Page 8, Lines 218-227, and Page 16, Lines 525-532.

Another key innovation of this framework lies in its ability to calculate the interaction correlations between drug combinations, allowing for the distinction of primary and secondary relationships, consistent with the principles of TCM. This approach has not been previously implemented or discussed in other studies.

Comments 5: Overall, the conclusion is written very sparsely. Considering the extensive nature of the paper, the conclusion could be much better formulated.

Response 5: We appreciate your valuable suggestions and apologize for the inadequate description in the previous conclusion section, particularly in summarizing the research innovation. In response to the reviewers' feedback, we have revised the conclusion to better highlight the functionality, importance, and applicability of the proposed research framework. By using gouty arthritis as an example, we demonstrate the framework's effectiveness in predicting drug combinations. The revised conclusion now emphasizes the comprehensiveness and innovation of our research while summarizing the key findings more clearly. The updated conclusion can be found on Page 19, Lines 637-651. Once again, we sincerely thank you for your insightful suggestions.

Response to Comments on the Quality of English Language

We deeply value the suggestions made by the reviewers regarding the manuscript's English language quality. As a result, we have carefully gone over and edited the language to improve readability, grammar, and clarity. We have made sure that the language used is in line with academic norms, which will help the text be read by a larger audience. If there are any particular portions that still need work, we're willing to make more changes to improve the language.

Additional clarifications

We appreciate the opportunity to address the reviewers' comments and provide additional clarifications. Here are some details that may help in understanding our revisions:

1. Sections Revisions: The abstract, introduction, methodology, findings, and discussion have all been extensively rewritten in this paper. These changes are intended to improve the text's consistency and readability while making sure that complicated ideas are communicated succinctly and clearly.

2. Model Description: We offer a more detailed explanation of the innovative aspects of the model's construction. Specifically, we demonstrate how the model enhances efficiency and generalizability by integrating molecular docking simulations with distance and overlap metrics, as well as standardizing clinical drug use. Additionally, we standardized the code environment to improve the model's adaptability for future applications.

3. Updates to Figures and Tables: We have optimized and updated all figures and tables.  To appropriately represent the data and bolster the conclusions of the article, every graphic component has undergone a reevaluation.

4. Language Quality: We chose a professional editor to polish the manuscript in order to guarantee that it satisfies the strict requirements needed for academic publication.

5. Additional Details and References: To support our views and give a stronger basis for our conclusions, we have added more information and references where appropriate.

6. Reaction to individual Comments: We have thoroughly answered each reviewer's individual comments, offering clarifications and arguments as needed. The updated manuscript and the attached response letter contain all the modifications and answers.

We hope these clarifications will assist in the review process and welcome any further feedback to enhance the quality of our work.

Round 2

Reviewer 1 Report

Comments and Suggestions for Authors

Thank you for the revised manuscript, it is already much better. Basically, I ask you to correct the pictures, the resolution on them still does not allow you to see the entire text in detail. After changing and replacing these illustrations, I consider the manuscript ready for publication!